# Spatiotemporal distribution of cutaneous leishmaniasis in Sri Lanka and future case burden estimates

**Nadira D. Karunaweera**[1‡]*, **Sanath Senanayake**[1], **Samitha Ginige**[2], **Hermali Silva**[1], **Nuwani Manamperi**[3], **Nilakshi Samaranayake**[1], **Rajika Dewasurendra**[1], **Panduka Karunanayake**[1], **Deepa Gamage**[2], **Nissanka de Silva**[4], **Upul Senarath**[1], **Guofa Zhou**[5‡]

**1** Faculty of Medicine, University of Colombo, Colombo, Sri Lanka, **2** Ministry of Health, Colombo, Sri Lanka, **3** Faculty of Medicine, University of Kelaniya, Ragama, Sri Lanka, **4** Faculty of Applied Sciences, University of Sri Jayewardenepura, Nugegoda, Sri Lanka, **5** University of California Irvine, Irvine, California, United States of America

‡ These authors were co-principal investigators.
* nadira@parasit.cmb.ac.lk

**Data Availability Statement:** All relevant data are within the manuscript and its Supporting Information files.

## Abstract

### Background

Leishmaniasis is a neglected tropical vector-borne disease, which is on the rise in Sri Lanka. Spatiotemporal and risk factor analyses are useful for understanding transmission dynamics, spatial clustering and predicting future disease distribution and trends to facilitate effective infection control.

### Methods

The nationwide clinically confirmed cutaneous leishmaniasis and climatic data were collected from 2001 to 2019. Hierarchical clustering and spatiotemporal cross-correlation analysis were used to measure the region-wide and local (between neighboring districts) synchrony of transmission. A mixed spatiotemporal regression-autoregression model was built to study the effects of climatic, neighboring-district dispersal, and infection carryover variables on leishmaniasis dynamics and spatial distribution. Same model without climatic variables was used to predict the future distribution and trends of leishmaniasis cases in Sri Lanka.

### Results

A total of 19,361 clinically confirmed leishmaniasis cases have been reported in Sri Lanka from 2001–2019. There were three phases identified: low-transmission phase (2001–2010), parasite population buildup phase (2011–2017), and outbreak phase (2018–2019). Spatially, the districts were divided into three groups based on similarity in temporal dynamics. The global mean correlation among district incidence dynamics was 0.30 (95% CI 0.25–0.35), and the localized mean correlation between neighboring districts was 0.58 (95% CI 0.42–0.73). Risk analysis for the seven districts with the highest incidence rates indicated

**Funding:** Research reported in this publication was supported by the National Institute of Allergy and Infectious Diseases, National Institutes of Health under award no. U01AI136033 to N.D.K. The authors are solely responsible for the content and it does not necessarily represent the official views of the National Institutes of Health. The funders had no role in study design, data collection and analysis, decision to publish or preparation of the manuscript.

**Competing interests:** The authors have declared that no competing interests exist.

that precipitation, neighboring-district effect, and infection carryover effect exhibited significant correlation with district-level incidence dynamics. Model-predicted incidence dynamics and case distribution matched well with observed results, except for the outbreak in 2018. The model-predicted 2020 case number is about 5,400 cases, with intensified transmission and expansion of high-transmission area. The predicted case number will be 9115 in 2022 and 19212 in 2025.

## Conclusions

The drastic upsurge in leishmaniasis cases in Sri Lanka in the last few year was unprecedented and it was strongly linked to precipitation, high burden of localized infections and inter-district dispersal. Targeted interventions are urgently needed to arrest an uncontrollable disease spread.

### Author summary

Leishmaniasis is on the rise in Sri Lanka in contrast to the declining trend in rest of South Asia. Spatiotemporal analysis and disease risk factors are useful for understanding transmission mechanisms and predicting future disease distribution to facilitate control. In this study we analyzed data on cutaneous leishmaniasis cases from Sri Lanka from 2001 to 2019. We asked three important questions regarding the driving forces behind the intensified leishmaniasis transmission: 1) Are the transmission dynamics in different areas synchronized? 2) What is the role of neighboring-area dispersal in shaping transmission dynamics? 3) How important is climatic variability in transmission dynamics? We used a multi-step approach to answer these questions. In addition to cross-correlation analysis, we built a mixed spatiotemporal regression-autoregression model to analyze risk factors, which is unique in leishmaniasis research because the simplified model was also useful for predicting future disease distribution. We found that the incidence dynamics in different districts could be divided into three synchronized groups based on similarity. Risk factor analysis indicated that precipitation, neighboring-district dispersal, and local infection carryover played important roles in shaping transmission dynamics. The spatiotemporal model predicted intensifying transmission with increasing case numbers, and expansion of high-transmission areas. Targeted interventions are urgently needed to stem the outbreak.

## Introduction

Leishmaniases are diseases caused by *Leishmania* spp. parasites transmitted through the bites of infected female sand flies [1]. There are three main types of leishmaniasis: visceral (VL), the most serious form of the disease; cutaneous (CL), the most common; and mucocutaneous (MCL), the most disabling form of the disease [1]. One hundred and two countries or territories remain endemic for leishmaniasis, with more than one billion people living at risk of acquiring the disease [1]. There are about 30,000 new cases of VL and >1 million new cases of CL that occur annually [1]. Although more than 100 countries are endemic to leishmaniases, high transmission is concentrated within a few. In 2018, over 90% of global VL cases were reported from seven countries: Brazil, Ethiopia, India, Kenya, Somalia, South Sudan, and

Sudan [1,2]. Over 90% of MCL cases occurred in Bolivia, Brazil, Ethiopia, and Peru, and 10 countries reported more than 5000 CL cases [1,2].

Three South Asian countries, India, Nepal, and Bangladesh, accounted for about half the global burden of VL [2]. Encouraged by the recent progress achieved in leishmaniasis control, these countries made a commitment to eliminate VL in the region by 2015 with the deadline extended thereafter [3,4]. The elimination target was set at less than one case per 10,000 people per year, an incidence rate considered to be no longer of public health concern [3,4]. However, Sri Lanka, India's neighbor, has reported a substantial surge in clinical leishmaniasis cases in the past 20 years [5]. The reported cases of leishmaniasis in Sri Lanka increased from 22 cases in 2001 to 426 cases in 2010 and 3,271 cases in 2018, with one district had an incidence rate of >10 cases/10,000 people/year and eight additional districts (out of the total 25 districts) had an incidence rate of >1 cases/10,000 people/year in 2018 [5]. The vast majority of leishmaniasis clinical cases in Sri Lanka are CL caused by *Leishmania donovani*, the same parasite that causes VL elsewhere, including in India [2,6]. The probable vector of the Sri Lankan leishmaniasis parasite is the *Phlebotomus glaucus* sand fly, which demonstrates zoophilic behavior and differs from the *Phlebotomus* species of sand flies found in southern India, near the India–Sri Lanka strait [7,8]. The continuous upsurge of the disease transmission in Sri Lanka may hamper the regional effort of eliminating VL in the South Asia and calls for local interventions.

The question is where to start the interventions? Intervention strategies should be built based upon the understanding of the epidemiology of leishmaniasis. Leishmaniasis studies in Sri Lanka have demonstrated two transmission hotspots, one in the south coast and another in the north central region of the country, with a possible biannual seasonal variation [5,9]. Clinical leishmaniasis in Sri Lanka also showed a shift in sex and age distribution, from predominantly males aged 21–40 in early 2000s to a more balanced sex ratio and a 50/50 split between ages 21–40 and >40 years since 2015 [10]. Furthermore, outdoor-associated activities, including occupational exposure and living near a potential vector breeding site, were identified as key risk factors of leishmaniasis infection in Sri Lanka [11,12]. Previous studies have also explored the spatiotemporal patterns of leishmaniasis infections [13–16], but most of them, if not all, lack predictive power, with the output limited to distribution patterns. Other studies used agent-based modeling to predict the distribution of leishmaniasis [17,18]. These models were built based on the deeper understanding of the biology of sand fly, parasite, and host [19] and therefore, useful for the understanding of the interactions among vectors, parasites, and hosts, and how environmental (include climatic) and socioeconomic (include demographic) factors influence the distribution of leishmaniasis infections [17–19]. Since some of the variables used in the models such as climatic data is only available in retrospect, these models are good for static risk assessments and suffer similar limitations of lack of predictive power for future distributions. Since long-term leishmaniasis surveillance data are available at local scale in many countries, this may provide a good opportunity to better utilize these clinical data for future transmission predictions [5,20–22]. Time series models are useful tools for analyzing such long-term observational data [23–25], however that too is devoid of a spatial component. Nonetheless, the information revealed through past studies provide a good understanding of leishmaniasis epidemiology and transmission and pave the way for greater public awareness and disease control [5,19]. However, further questions remain regarding the analyses and predictions of spatiotemporal dynamics in leishmaniasis transmission and the driving forces behind it. For example, is the leishmaniasis parasite circulated locally or spread between neighboring areas? This information is important for determining control strategies. Furthermore, since the development of both the sand fly and the parasite inside its gut are affected by climatic conditions [26,27]; how do climatic factors affect leishmaniasis transmission? How can we better utilize this information to predict the future distribution of leishmaniasis in Sri

Lanka? To answer these questions, spatial-temporal predictive models are required, which can predict the influence of potential parasite movement between neighboring areas that can facilitate spread. However, the persistence of local transmission may also lead to a buildup of case numbers. These components remain as gaps from previous work in this field [13–19,23–25].

The aim of this study was to further explore spatiotemporal leishmaniasis transmission patterns; to examine the contributions of climatic factors, local carryover of infections, and inter-region transmission to an epidemic situation; and to develop space-time models for the prediction of future clinical case distribution in Sri Lanka. This study can serve as a prototype of an early warning model for the prediction of future epidemics.

## Methods

### Ethics statement

Ethics approval for the study was obtained from the ethics review committee, Faculty of Medicine, University of Colombo (Reference number EC-17-062). This study only used aggregated data collected from health facilities, without any individual patient information, therefore, no consent/assent were required.

### Study area

This study covered the entire Sri Lanka. It's a country located in South Asia on the southeastern tip of the Indian sub-continent between latitudes 5° and 10° N, and longitudes 79° and 82° E. It is an island with an area of 65,610 km$^2$ and a population of 21,803,000 (2019 estimate; 2012 census population 20,277,597) (http://www.statistics.gov.lk). Administratively, it has 25 districts and 357 divisions (vary over time due to the creation of new divisions). It consists mostly of flat to rolling coastal plains, with mountains rising in the south-central part with the highest elevation of 2,524 meters above sea level. The climate in Sri Lanka is tropical and warm with a mean temperature ranging from 17°C in the highlands to 33°C in the coastal areas. The rainfall patterns are influenced by the seasonal monsoon winds, with an annual rainfall of 2,500 mm in the central highlands down to around 1,200 mm in some of the dry zones of east, southeast, and northern parts of Sri Lanka. The annual rainfall is from 800–1,200 mm in the semiarid northwest and southeast coasts.

### Clinical and climatic data collection

This study focused only on anthroponotic cutaneous leishmaniasis. Clinical data collection has been described in a previous study [5]. Briefly, clinically-confirmed leishmaniasis case data was obtained from the records maintained at the diagnostic and research laboratory at the Faculty of Medicine, University of Colombo, the Epidemiology Unit of the Ministry of Health, Sri Lanka and from small health administrative units (the medical officer of health branches) in each district. Case counts of CL at each district covered the entire country from 2001 to 2019; division-level case counts covered the period from 2015 to 2019. The population in each district was obtained from census data of 2001 and 2012 with projections for each year as given by the government of Sri Lanka, the population in each division was obtained from 2012 Sri Lanka census (http://www.statistics.gov.lk). For the district level data analysis, annual total population was used to calculate the disease incidence rate in a given district and given year from 2001 to 2019. For the division level data analysis, 2012 census projected population at each division was used to calculate incidence rate at division level for years from 2015 to 2019. The climatic data was collected from 20 meteorological stations where long-term climatic data are available in the country. Climatic variables included monthly mean of maximum,

minimum, and mean temperature and monthly accumulative precipitation from 2001 to 2019. Climatic data was only used for district level disease data analysis.

## Spatial distribution and temporal trend classification

The distribution of district-level annual incidence rates was used to determine the temporal trend in leishmaniasis transmission, which was analyzed using hierarchical clustering to determine the similarity in incidence rates among different years (spatial similarity) [28]. The purpose of this step was to analyze how disease transmission had evolved over the past 19 years. The district-level incidence dynamics and hierarchical clustering was used to determine the similarity in incidence dynamics among different districts (temporal similarity) [28]. The last step was to measure space-time correlation, i.e., to determine transmission spatial synchrony [29]. Spatial synchrony refers to coincident changes in the incidence or other time-varying characteristics of geographically adjacent areas [29]. Ecologically, disease spatial synchrony may arise from three primary mechanisms, i.e., dispersal among areas, congruent dependence of incidence dynamics on exogenous factors such as climatic parameters, and host-vector-parasite interactions in which vectors are spatially synchronous [29]. This analysis will help to explain the mechanisms behind disease transmission, e.g., spatial dispersal, locally correlated climatic variables, and/or how vectors might interact with local parasites to produce different patterns of transmission. The cross-correlation coefficient was used to measure region-wide transmission synchrony, and a modified Mantel correlogram to measure local transmission synchrony [29,30]. The correlations were only measured between neighboring districts and correlation with farther distance was not measured due to limited number of districts.

## Risk factor analysis

This step was included to identify the driving forces behind transmission dynamics in different districts. Three types of risk factors were tested in this study: climatic variables, inter-district dispersal, and localized transmission. The following mixed regression-autoregression model incorporated with a spatial component was used to assess the risk factors [31–33]:

$$I(D_i, t+1) = \alpha + \beta * I(D_i, t) + \gamma * \sum_{ND} I(ND, t) + f(climate, i)$$

$$f(climate, i) = \beta_1 MaxT(D_i, t) + \beta_2 MeanT(D_i, t) + \beta_3 MinT(D_i, t) + \beta_4 Precip(D_i, t)$$

Where $I(D_i, t)$ stands for incidence rate at district $i$ in year $t$; $ND$ is all neighboring districts of district $i$; $\Sigma$ is the sum of all incidence in neighboring districts; $MaxT$, $MeanT$, $MinT$ and $Precip$ represent the annual average maximum, mean, and minimum temperature and the annual cumulative precipitation at district $i$, respectively; and $\alpha$, $\beta$, $\gamma$, and $\beta_i$, i = 1–4, are constant coefficients.

The climatic effect was tested to measure if the dynamics of infections were affected by local climatic factors. The neighboring-district effect was tested to determine if the infections in a given district were associated with the dispersal of parasites from neighboring districts. Infections from the previous year were used to test the local infection carryover effect, i.e., the localized transmission trend carried over from the previous year. Here the carryover effect means the effect of the current infections in a given area on the future infections in the same area. In other words, if a district has high number of confirmed infections this year, one would expect a similarly high number of infections in that district by next year because parasites may well be maintained within local vectors or other reservoir host populations. Risk factor analysis was

conducted only in districts where mean annual incidence rate > 5 cases/1,000 population and where climatic observations were available.

The risk factors were tested in an additive fashion; i.e., we first tested the climatic effect, with subsequent additions of the neighboring effect, and thereafter, the carryover effect. The stepwise multiple regression analysis and least square method for variable selection were used to determine if one factor was more significantly correlated with leishmaniasis incidence dynamics than the others. The correlation coefficient was used to measure the overall goodness-of-fit of the model.

### Space-time prediction

If localized spatiotemporal correlation existed as determined from the previous two steps, and if the climatic effect was not strong, it was possible to predict the future case distribution based on the carryover and neighboring-district dispersal effects. While this approach may underutilize the climatic information, the model can predict future case numbers without climatic data (since future climatic data is unavailable) to make predictions. Since local transmission and spatial dispersal may occur at a finer scale, we used division-level data to build the following predictive model [33,34]:

$$I(D_i, t+1) = \alpha + \beta * I(D_i, t) + \gamma * \sum_{ND} I(ND, t)$$

Where, $I(D_i, t)$ is the number of cases in division $i$ at time $t$; $ND$ represents neighboring divisions of division $i$; and $\alpha$, $\beta$, and $\gamma$ are constant coefficients. The least-square multiple regression analysis was used for model parameter estimation. However, directly observed climatic data at division level was not available for use in Sri Lanka.

The division-level case numbers from 2015–2018 were used to build the model and 2019 data was used for validation. We used the model to predict the case distribution for 2020 and total number of cases from 2020 up to 2025 together with their 95% confidence intervals.

The maps and spatial data were generated using ArcGIS 10.0 (Redlands, CA, USA). Other data analyses were conducted using R 3.6.3. with four R packages, viz. car, lme4, MASS, and synchrony.

## Results

### Leishmaniasis epidemiology: Spatiotemporal dynamics

There were 19,361 clinically-confirmed leishmaniasis cases from 2001 to 2019. Case incidences were low from 2001 to 2008, followed by a slow but steady increase in case incidence from 2009 to 2017, until a sudden outbreak occurred in 2018 (Fig 1A) with case numbers increasing from 1,508 cases in 2017 to 3,271 in 2018, and then 4,061 cases in 2019. District-level average incidence rate increased about 4-fold from 2017 to 2018. Four districts had average annual incidence rates ≥ 10 cases/1,000 people, and another nine districts had annual incidence rates ≥ 1 case/1,000 people (Fig 1B and 1C).

Classification of district-level incidence distribution found that the study period could be classified into three based on similar distribution patterns: 2001–2010, 2011–2017, and 2018–2019 (Fig 2). All average incidence rates were < 1 case/1,000 people from 2001 to 2010, between 1 and 5 from 2011 to 2017, and >10 cases/1,000 people after 2017 (Fig 2). The similarity in incidence distributions did not follow chronological order, as indicated in branches of the classification tree (Fig 2). Overall, 2001–2010 could be seen as a low-transmission period, 2011–2017 as a parasite population build-up period, and 2018–2019 as an epidemic period (Figs 1A and 2).

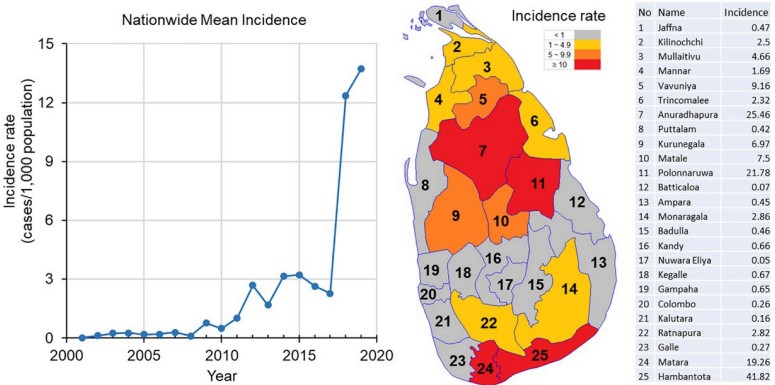

**Fig 1.** a) Nationwide mean annual incidence rate (cases/1,000 people/year) from 2001 to 2019 (left), b) the distribution of average incidence rate in each district over the study period (middle), and c) the corresponding names and incidence rate in each district (right). Source: Authors' own map using data from https://data.humdata.org/dataset/sri-lanka-administrative-levels-0-4-boundaries.

Districts could be classified into three groups based on similarity in incidence dynamics (Fig 3). The majority of the districts with low incidence rates were in one group (gray on the map, Fig 3). Four districts with intermediate incidence rates had similar temporal dynamics (yellow on the map). The remaining seven districts (green on the map) were not necessarily classified into one group; however, they had temporal dynamics that differed from the other two groups, and they were also the seven districts with the highest average annual incidence rates (Fig 3). Incidence rates in these seven districts were subjected to further risk factor analysis.

The heat map in Fig 4 illustrates the distribution of incidence rates over time (Fig 4). Results from the space-time joining classification indicated that the study areas could be roughly classified into three groups: A) continuously low incidence until 2019 (top section); B) relatively high incidence before 2008, followed by a decline and then a resurgence starting in 2012 (middle section); and C) continuously increasing incidence (bottom section) (Fig 4).

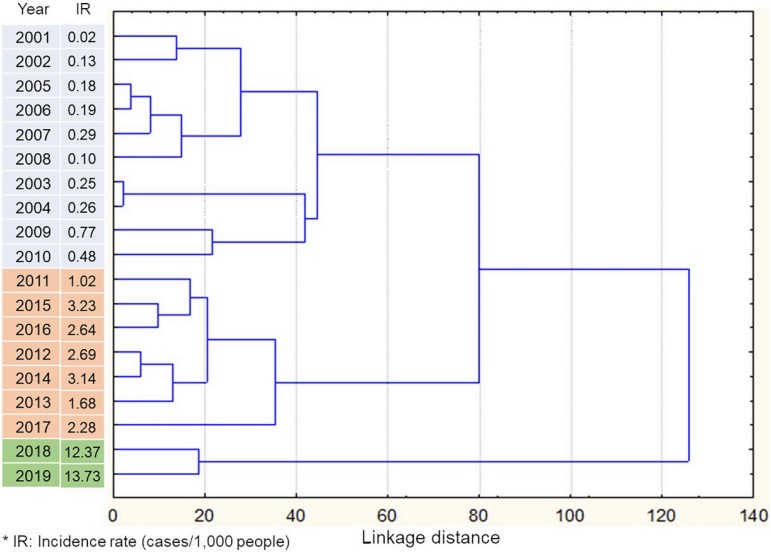

**Fig 2. Classification of study period by disease distribution at each district.**

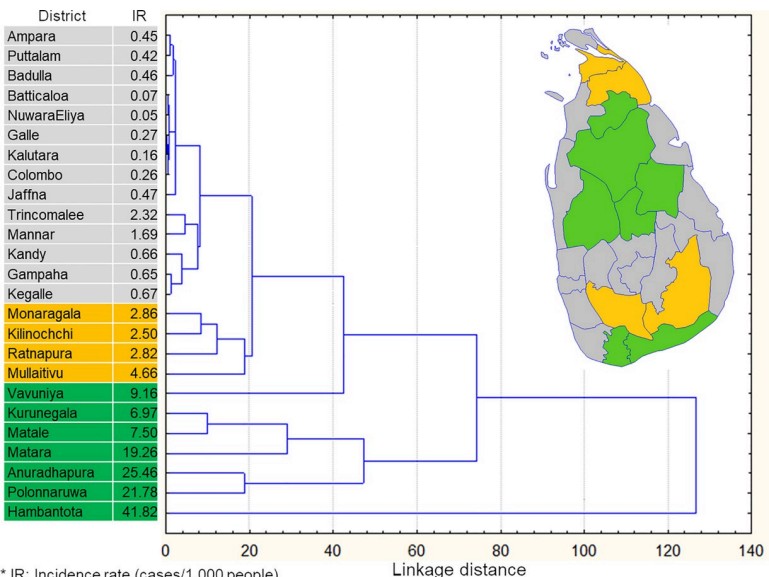

**Fig 3. Classification of study areas by disease dynamics at each district.** Source: Authors' own map using data from https://data.humdata.org/dataset/sri-lanka-administrative-levels-0-4-boundaries.

The global mean correlation among district incidence dynamics was 0.30 (95% CI 0.25–0.35). However, the localized Mantel correlation between neighboring districts was on average 0.58 (95% CI 0.42–0.73), indicating a rather high correlation between neighboring districts.

## Risk factor analysis

The seven districts for which we built risk analysis models were all classified into one group based on transmission dynamics (Fig 3), and they all had average annual incidence rates > 5 cases/1,000 people. The climatic variables–only model showed that three out of seven models yielded a correlation coefficient $R^2 > 0.5$, indicating good correlation with climatic factors ($R^2$ values 0.711–0.829), while two districts showed no significant correlation between disease

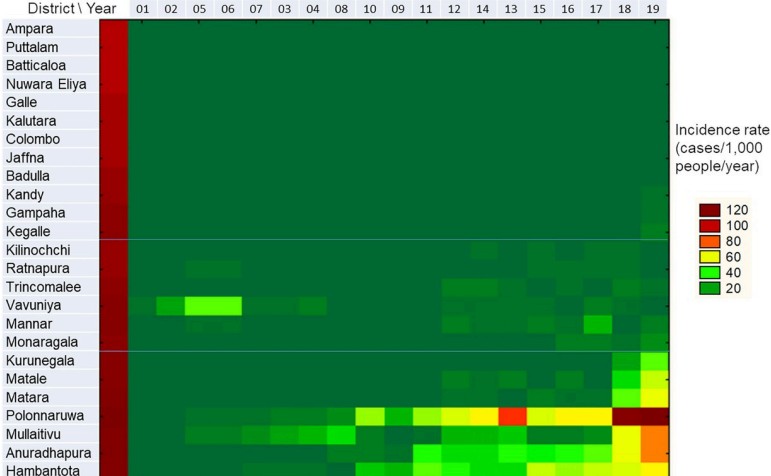

**Fig 4. Joining classification of study areas and study period by incidence rate at each district.**

**Table 1. Risk factors of clinical leishmaniasis in different districts.**

| District | Hambantota | Matara | Kurunegala | Polonnaruwa | Matale | Anuradhapura | Vavuniya |
|---|---|---|---|---|---|---|---|
| Mean annual incidence rate (cases/ 1,000 population) | 41.817 | 19.264 | 6.972 | 21.779 | 7.505 | 23.532 | 9.550 |
| Climate model | | | | | | | |
| Adjusted $R^2$ | 0.829 | N/A | N/A | 0.784 | 0.425 | 0.711 | 0.185 |
| ANOVA F- value | 30.127 | | | 22.812 | 7.648 | 23.196 | 5.079 |
| P-value | <0.0001 | | | <0.0001 | 0.0047 | <0.0001 | 0.0377 |
| Coefficients | | | | | | | |
| Minimum temperature | -68.499*** | | | | | -78.453*** | 6.547* |
| Mean temperature | 64.465 ** | | | 83.635*** | 64.183** | | |
| Maximum temperature | -35.656 * | | | -37.536 ** | -39.912 * | 69.320*** | |
| Precipitation | | | | | | | |
| Climate + Neighbor-district model | | | | | | | |
| Adjusted $R^2$ | 0.838 | 0.808 | 0.873 | 0.930 | 0.832 | 0.726 | 0.185 |
| ANOVA F- value | 47.675 | 38.967 | 62.606 | 119.738 | 45.612 | 24.861 | 5.079 |
| P value | <0.0001 | <0.0001 | <0.0001 | <0.0001 | <0.0001 | <0.0001 | 0.0377 |
| Coefficients | | | | | | | |
| Minimum temperature | -35.213 ** | 22.311*** | | | | | 6.547* |
| Mean temperature | | | | | | | |
| Maximum temperature | | | | | | | |
| Precipitation | | | -0.005 ** | 0.004 * | -0.009*** | 0.014 ** | |
| Neighboring-district mean | 3.340 *** | 1.262*** | 1.815*** | 3.061*** | 1.551*** | 2.444*** | |
| Climate + Neighbor-district+Carryover model | | | | | | | |
| Adjusted $R^2$ | 0.915 | 0.846 | 0.937 | 0.930 | 0.832 | 0.816 | 0.316 |
| ANOVA F- value | 65.449 | 33.858 | 89.557 | 119.738 | 45.612 | 40.796 | 9.310 |
| P-value | <0.0001 | <0.0001 | <0.0001 | <0.0001 | <0.0001 | <0.0001 | 0.0072 |
| Coefficients | | | | | | | |
| Minimum temperature | | | | | | | |
| Mean temperature | | | | | | | |
| Maximum temperature | | | | | | | |
| Precipitation | 0.015* | -0.009* | -0.003* | 0.004* | -0.009* | | |

Note: Stepwise regression analysis was used, insignificant variables were removed at level of 0.05

*, ** and*** represent significant level of <0.05, <0.01 and <0.001, respectively.

dynamics and temperature or precipitation variables (Table 1). Without considering the neighboring-district dispersal and carryover effects, precipitation alone was not significantly correlated (P < 0.05) with leishmaniasis transmission dynamics (Table 1).

When the neighboring-district effect was included, six of the seven models had $R^2 > 0.7$ ($R^2$ values 0.726–0.930) and the neighboring-district effect appeared in six models (P < 0.05), precipitation appeared in four models (P < 0.05), and one model still showed very weak correlation (Vavuniya District, $R^2 = 0.18$) between transmission dynamics and risk factors (Table 1).

When climatic, neighboring-district dispersal, and carryover effects were all included, precipitation appeared in five models, neighboring-district effect in six models, and carryover effect in five models ($R^2$ values 0.816–0.930). Temperature did not appear in any of the models tested (Table 1). Once again, models for disease dynamics in Vavuniya District showed weak correlation ($R^2 = 0.32$) with the risk factors investigated; only the carryover effect was selected by the stepwise process, indicating probable locally isolated transmission (Table 1).

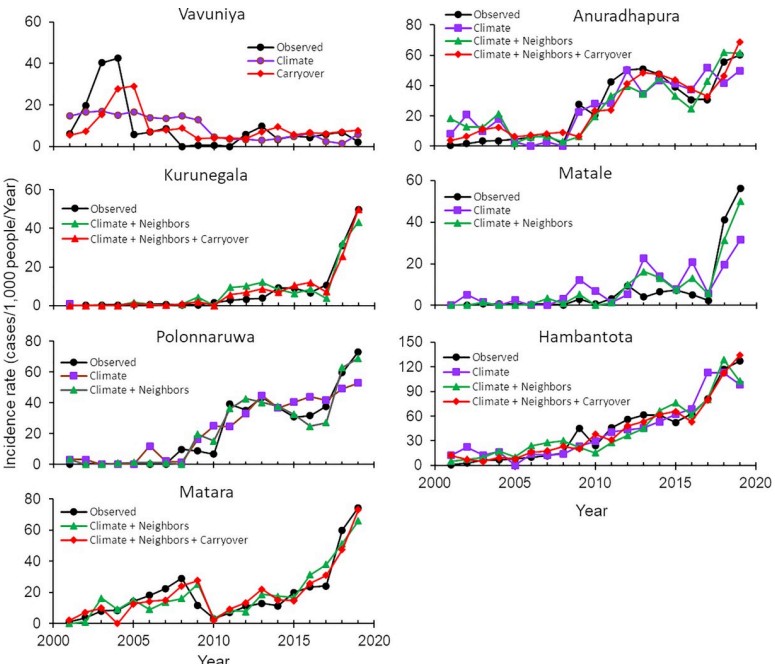

**Fig 5. Observed and model-predicted temporal changes in incidence rate in the seven districts with the highest case numbers.** Climate, neighbor, and carryover represent climatic, neighboring-district dispersal, and carryover effects, respectively. The results of goodness-of-fit of each model were presented in Table 1, predictions was only made with significantly fitted models.

Overall, the risk model–predicted disease dynamics matched well with the observed dynamics in six of the seven districts, except Vavuniya (Fig 5 and Table 1). Leishmaniasis transmission in Vavuniya was likely to be an outlier.

## Spatial distribution and future trend prediction

Using cases from the previous year in the same division and in neighboring divisions, we built a prediction model with $R^2 = 0.823$ and adjusted $R^2 = 0.823$ (ANOVA $F_{2,1321} = 3075.71$, $P < 0.0001$, Table 2). Model-predicted cases matched well with observed cases for all years except 2018, which was an epidemic year (Figs 6 and 7 and S1). The key mismatch between observed and predicted cases in 2018 was the intensity in the north-central and southwestern coastal areas, where the model underestimated the observations in some high-transmission divisions (Figs 6 and S1). The model predicted both the expansion of the epidemic area and the increased transmission intensity, as well as the expansion of the southern hotspot from coastal to inland areas (Fig 6). The predicted case number in 2020 is 5,379 (95% CI [4739, 6017]), which is about a 30% increase from 2019 observations (4,064 cases) (Fig 7). If the current trend holds, the predicted case number will be 9,115 (95% CI [8,157, 10,070]) by year 2022 and close to 20,000 by year 2025 (19,212 cases, 95% CI [17,389, 21,31]).

**Table 2. Division-level predictive model summary.**

| Variable | Coefficient (95% CI) | t-value | p value |
|---|---|---|---|
| Intercept | 0.8102 [0.1921, 1.4284] | 2.57 | 0.0102 |
| Cases from previous year | 1.0679 [1.0273, 1.1086] | 51.52 | < 0.0001 |
| Cases in neighboring divisions from previous year | 0.0331 [0.0215, 0.0446] | 5.61 | < 0.0001 |

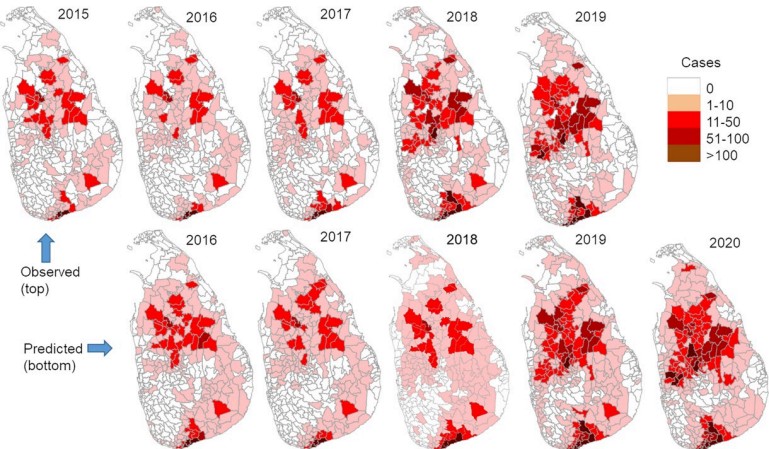

**Fig 6. Observed and model-predicted leishmaniasis cases by division and year.** Source: Authors' own map using data from https://data.humdata.org/dataset/sri-lanka-administrative-levels-0-4-boundaries.

## Discussion

Leishmaniasis clinical cases are still on the rise as reported by the Sri Lankan Ministry of Health, in contrast to the substantial decline in case numbers recorded in other South Asian countries including India, Sri Lanka's nearest neighbor [2,5]. Although leishmaniasis clinical cases were first reported from Sri Lanka in 1992 [35], the case numbers have remained low and sporadic prior to 2001 [36]. Reported case numbers increased 15-fold from 2001 to 2010, followed by a 10-fold increase from 2010 to 2019, with a clear outbreak in 2017–2018. In addition to localized transmission, inter-area dispersal seems to have played an important role in spreading the parasite, and precipitation is probably the key climatic factor fueling the increasing trends in infections. The spatiotemporal model predicted that cases will continue to rise unless viable interventions are implemented. There are no formal leishmaniasis interventions implemented either locally or nationwide in Sri Lanka, except for diagnosis and treatment of patients who report to hospitals. With due consideration given to the highly localized transmission with spread among neighboring areas, introduction of focal vector control measures may help to effectively reduce the local transmission and parasite spread, since factors such as

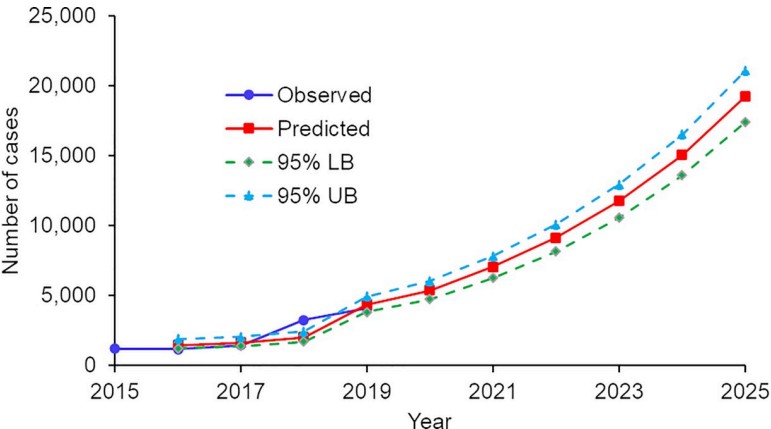

**Fig 7. Observed vs. model-predicted total cases from 2015 to 2025.**

the climate, population movement and vector dispersal are beyond control., The potential risk of uncontrollable epidemic calls for urgent introduction of effective intervention strategies.

Spatiotemporal analysis is a powerful tool, not only for detecting transmission hotspots but also for understanding the driving forces behind transmission, e.g., how spatial dispersal and locally correlated climatic variables contribute to transmission [11,28,29,37–39]. The spatio-temporal and risk factor analyses of epidemiological studies conducted in the past have yielded limited outcome [11,37–42] and they lacked predictive power [43,44] and failed to optimally utilize spatial information [20–22]. The multi-step approach adopted in this study included not only spatiotemporal clustering but also spatiotemporal risk modeling. Study of disease dynamics in Sri Lanka based on spatiotemporal classification analysis described in this paper demonstrated that the study period from 2001 to 2019 may be divided into three phases: the low-transmission phase from 2001 to 2010, the parasite population build-up phase from 2011 to 2017, and the sudden epidemic starting in 2018. From 2001 to 2010, parasites circulated locally with a few hotspots [5]. Prior to 2017, no more than three districts had relatively high incidence rates (>50 cases/1,000 people-years) each year. However, this number increased to five districts in 2018 and seven in 2019. Spatial correlation analysis indicated that transmission dynamics between neighboring districts were positively correlated, suggesting inter-area spreading. Risk factor analysis in the heavily hit districts suggested that precipitation was likely the key climatic factor driving the increase in cases. Persisting CL lesions due to poor respon-siveness to routine therapy [45] may have contributed to the sudden leishmaniasis surge in Sri Lanka in 2018. However, further investigations will be required for its confirmation. The results from this study provide a better understanding of the epidemiology of leishmaniasis and the factors related to the emergence of the disease. The temporal prediction is useful for epidemic preparedness, and the spatial prediction can help to guide targeted interventions for control.

Leishmaniasis transmission in Vavuniya District was a special case. Vavuniya and Kilinoch-chi districts were in the front lines of the 30-year armed conflict in Sri Lanka, which ended in 2009 [45,46]. With the existence of indigenous circulating leishmaniasis parasites, the trans-mission of leishmaniasis in Vavuniya District seemed focal, and might be due to the displace-ment of locals during the conflict and the postwar return of individuals who had fled, coupled with the influx of armed forces personnel who assisted with resettlement and reconstruction of infrastructure [46,47]. It may be described as a test cage, where parasites circulated freely. Once resettlement was completed, local infections may have been mass tested, clinical cases treated and armed forces personnel redeployed elsewhere [46,47], which may have influenced the control of parasite transmission. Such factors may explain the failure to reliably predict the transmission dynamics in this district in relation to the climatic variables and neighboring-dis-trict dispersal.

A limitation of this study is the spatiotemporal predictive model, which lacked the power to predict the 2018 leishmaniasis outbreak. Since long-term dynamics of climatic variables can-not be predicted in a futuristic manner, any climate-based model will lack predictive power. Climatic factors were not included in the predictive model in this study; however, if climatic factors were indeed the driving force behind the 2018 outbreak, model predictions excluding climatic variables would surely miss the epidemic. Studies in other countries have linked cli-matic variability with outbreaks of leishmaniasis [23–25]. Using long-term mean climatic data to predict future disease transmission is an option. However, it lacks variability and thus likely to miss the abnormal changes; in addition, it creates another dimension of uncertainty for the predictions. Nonetheless, we speculate that climatic variation and/or persistent CL lesions due to poor drug response may have been the causes of the 2018 outbreak in Sri Lanka; however, further evidence is required. Another limitation is the short temporal period used for building

the predictive model (the model without climatic variables). We collected disease data from some 350 divisions for each year, which provided a good sample size in terms of the goodness-of-fit of the model; still, four years is a short period. Adding data observed from other years might improve model performance, although collecting accurate long-term nationwide division-level clinical case data has proven difficult. There are other minor limitations in this study, for example, the neighboring area effect and annual climatic data. Since monthly confirmed case numbers in each district are available since 2014, using monthly climatic data may fine tune the modeling results at least to a certain degree. At divisional level, infection may spread beyond the neighboring divisions because some divisions are small in size. Although the second-order neighborhood or distance weighted neighborhood effect could have been analyzed, this may have diverged the focus of the study.

In conclusion, leishmaniasis transmission in Sri Lanka is still spreading and has recently intensified. Spatiotemporal analysis incorporating a neighboring-area dispersal effect provides a new framework for analyzing the spatiotemporal dynamics of the disease, detecting the risk factors that affect the dynamics, and predicting potential future distribution of infections. Such information on future transmission distribution predictions are useful for disease control programs and policy makers for planning and prioritizing of targeted areas.

## Supporting information

**S1 Fig. Correlation between observed and predicted cases at each division from 2016 to 2019.**
(TIF)

## Acknowledgments

We are grateful to the head and staff of the Department of Parasitology, Faculty of Medicine, University of Colombo, and the Director General and staff of the Ministry of Health, especially the medical health officers, other health officials, and epidemiologists, for assistance in this study. We thank Raushan Siraj, Sachee Bhanu, Nishanthan Chandrasekeran, Dasun Dissanayake and Ruksala Ranatunga for assistance with data collection. This study was performed in the Faculty of Medicine of the University of Colombo, Sri Lanka.

## Author Contributions

**Conceptualization:** Nadira D. Karunaweera, Guofa Zhou.

**Data curation:** Guofa Zhou.

**Formal analysis:** Guofa Zhou.

**Funding acquisition:** Nadira D. Karunaweera.

**Investigation:** Nadira D. Karunaweera, Sanath Senanayake, Samitha Ginige, Hermali Silva, Nuwani Manamperi, Nilakshi Samaranayake, Rajika Dewasurendra, Panduka Karunanayake, Deepa Gamage, Nissanka de Silva, Upul Senarath.

**Methodology:** Nadira D. Karunaweera, Sanath Senanayake, Samitha Ginige, Hermali Silva, Nuwani Manamperi, Nilakshi Samaranayake, Rajika Dewasurendra, Panduka Karunanayake, Deepa Gamage, Nissanka de Silva, Upul Senarath, Guofa Zhou.

**Project administration:** Nadira D. Karunaweera.

**Resources:** Nadira D. Karunaweera.

**Software:** Guofa Zhou.

**Supervision:** Nadira D. Karunaweera.

**Validation:** Nadira D. Karunaweera, Guofa Zhou.

**Visualization:** Nadira D. Karunaweera.

**Writing – original draft:** Nadira D. Karunaweera, Guofa Zhou.

**Writing – review & editing:** Nadira D. Karunaweera.

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
