## [Decision Letter · Decision Letter 0]

2 Feb 2021

Dear Prof. Karunaweera,

Thank you very much for submitting your manuscript "Spatiotemporal transmission of cutaneous leishmaniasis in Sri Lanka and future case burden estimates" for consideration at PLOS Neglected Tropical Diseases. As with all papers reviewed by the journal, your manuscript was reviewed by members of the editorial board and by several independent reviewers. In light of the reviews (below this email), we would like to invite the resubmission of a significantly-revised version that takes into account the reviewers' comments. 

We cannot make any decision about publication until we have seen the revised manuscript and your response to the reviewers' comments. Your revised manuscript is also likely to be sent to reviewers for further evaluation.

Sincerely,

Alberto Novaes Ramos Jr

Associate Editor

Epco Hasker

Deputy Editor

Reviewer's Responses to Questions

**Key Review Criteria Required for Acceptance?**

**Methods**

-Are the objectives of the study clearly articulated with a clear testable hypothesis stated?

-Is the study design appropriate to address the stated objectives?

-Is the population clearly described and appropriate for the hypothesis being tested?

-Is the sample size sufficient to ensure adequate power to address the hypothesis being tested?

-Were correct statistical analysis used to support conclusions?

-Are there concerns about ethical or regulatory requirements being met?

Reviewer #1: Yes

Reviewer #2: my comments are attached.

Reviewer #3: Partially.

Reviewer #4: Major Revision

Reviewer #5: Please mention from where did you obtain meteorological data.

How many clinically confirmed cases you included for this study?

What are the tests you applied to analyze risk factors

**Results**

-Does the analysis presented match the analysis plan?

-Are the results clearly and completely presented?

-Are the figures (Tables, Images) of sufficient quality for clarity?

Reviewer #1: Yes

Reviewer #2: my comments are attached.

Reviewer #3: Yes

Reviewer #4: Minor Revision

Reviewer #5: fig 1 (b) incidence rate - which year? ?

Add p values for each clinical factors in every districts

**Conclusions**

-Are the conclusions supported by the data presented?

-Are the limitations of analysis clearly described?

-Do the authors discuss how these data can be helpful to advance our understanding of the topic under study?

-Is public health relevance addressed?

Reviewer #1: Yes

Reviewer #2: my comments are attached.

Reviewer #3: Yes

Reviewer #4: Major Revision

Reviewer #5: conclusions supported by the data presented

limitations of analysis clearly described

**Editorial and Data Presentation Modifications?**

Reviewer #1: Minor Revision

Reviewer #2: my comments are attached.

Reviewer #3: (No Response)

Reviewer #4: (No Response)

Reviewer #5: (No Response)

**Summary and General Comments**

Reviewer #1: In this study the authors addressed spatiotemporal transmission of cutaneous leishmaniasis and predicted future case burden in Sri Lanka. The objective of this study is clear, the data are well-analyzed, and the results are well-presented. This analytic model will be a powerful tool for the prediction of future risk of leishmaniasis, and contribute to the control of this disease I enjoyed reading this manuscript. Following is a minor comment.

1. The data of figure 5 is not explained well in the text. I recommend mentioning about it more detail in the text, or move this figure into a supplemental figure.

Reviewer #2: my comments are attached.

Reviewer #3: The paper ‘Spatiotemporal transmission of cutaneous leishmaniasis in Sri Lanka and future case burden estimates’ covers a very relevant and urgent topic in the Sri Lank context.

Below you can find recommendations to further improve this investigation:

L99 – Please provide a detailed information about this upsurge in leishmaniasis cases. This is critical to justify the need for the present study.

L128-138 – The sections about the study area and data sources is very incomplete. The following information needs to be included:

a. Description of the territorial units of analysis, more precisely population size (mean and SD or median and IQR)

b. Description of the information systems that collect and provide data on leishmaniosis cases (flow of information from diagnosis until register in the database; coverage; existence of changes in data quality through time, etc.)

c. Description of the climatic data, more precisely the data source, temporal resolution and number of meteorological stations in the climate network from Sri Lanka.

If data sources are prone to any type of information and selection bias, that needs to be discussed in the limitations section of the manuscript.

L148 – Please clarify the concept of transmission synchrony. The readers may not know this term.

L190 – Please indicate the R packages used for the analysis.

L180 – The authors do not use climatic variable to make the space-time predictions. However, the authors found that climatic factors played a role in the infections. Could it be possible to make predictions based on meteorological scenarios (high, low, medium precipitation year; high, low, medium temperature year)? I believe it would be an interesting add-on to the present or future research.

Figure 1 – To make the figure more informative I recommend numbering the districts and add the names as a legend of the figure.

Reviewer #4: (No Response)

Reviewer #5: Don't start sentences as "we.................. Please write in a scientific way.

Please give your suggestions more about how a high correlation between neighboring districts and incidence rate of leishamaniasis in endemic regions.

PLOS authors have the option to publish the peer review history of their article (what does this mean?). If published, this will include your full peer review and any attached files.

Reviewer #1: No

Reviewer #2: No

Reviewer #3: Yes: Ana Isabel Ribeiro

Reviewer #4: No

Reviewer #5: Yes: Lahiru Sandaruwan Galgamuwa
---

## [Decision Letter · Decision Letter 1]

30 Mar 2021

Dear Prof. Karunaweera,

We are pleased to inform you that your manuscript 'Spatiotemporal distribution of cutaneous leishmaniasis in Sri Lanka and future case burden estimates' has been provisionally accepted for publication in PLOS Neglected Tropical Diseases.

Best regards,

Alberto Novaes Ramos Jr

Associate Editor

Epco Hasker

Deputy Editor

Reviewer's Responses to Questions

**Key Review Criteria Required for Acceptance?**

**Methods**

-Are the objectives of the study clearly articulated with a clear testable hypothesis stated?

-Is the study design appropriate to address the stated objectives?

-Is the population clearly described and appropriate for the hypothesis being tested?

-Is the sample size sufficient to ensure adequate power to address the hypothesis being tested?

-Were correct statistical analysis used to support conclusions?

-Are there concerns about ethical or regulatory requirements being met?

Reviewer #1: Yes

Reviewer #3: The authors addressed my previous concerns in the revision

Reviewer #4: Yes

Reviewer #5: yes

**Results**

-Does the analysis presented match the analysis plan?

-Are the results clearly and completely presented?

-Are the figures (Tables, Images) of sufficient quality for clarity?

Reviewer #1: Yes

Reviewer #3: The authors addressed my previous concerns in the revision

Reviewer #4: Yes

Reviewer #5: yes

**Conclusions**

-Are the conclusions supported by the data presented?

-Are the limitations of analysis clearly described?

-Do the authors discuss how these data can be helpful to advance our understanding of the topic under study?

-Is public health relevance addressed?

Reviewer #1: Yes

Reviewer #3: (No Response)

Reviewer #4: Yes

Reviewer #5: yes

**Editorial and Data Presentation Modifications?**

Reviewer #1: (No Response)

Reviewer #3: no

Reviewer #4: Accept

Reviewer #5: Accept

**Summary and General Comments**

Reviewer #1: The paper was revised properly.

Reviewer #3: The authors addressed my previous concerns in the revision

Reviewer #4: I have read with interest your manuscript as well as reviewers’ concerns and your response. The revisions that you made to the manuscript are very effective in addressing the remaining concerns. The revised manuscript is overall well-written, well-structured and clear. So, in my opinion, the manuscript is well-suited for publication in PLOS Neglected Tropical Diseases.

Reviewer #5: Accept

PLOS authors have the option to publish the peer review history of their article (what does this mean?). If published, this will include your full peer review and any attached files.

Reviewer #1: No

Reviewer #3: **Yes: **Ana Isabel Ribeiro

Reviewer #4: No

Reviewer #5: **Yes: **Lahiru Sandaruwan Galgamuwa

---

## [Editor Report · Acceptance letter]

14 Apr 2021

Dear Prof. Karunaweera,

We are delighted to inform you that your manuscript, "Spatiotemporal distribution of cutaneous leishmaniasis in Sri Lanka and future case burden estimates," has been formally accepted for publication in PLOS Neglected Tropical Diseases.

Best regards,

Shaden Kamhawi

co-Editor-in-Chief

Paul Brindley

co-Editor-in-Chief
